# Protein kinase C coordinates histone H3 phosphorylation and acetylation

**Zoulfia Darieva, Aaron Webber, Stacey Warwood, Andrew D Sharrocks\***

Faculty of Life Sciences, University of Manchester, Manchester, United Kingdom

**Abstract** The re-assembly of chromatin following DNA replication is a critical event in the maintenance of genome integrity. Histone H3 acetylation at K56 and phosphorylation at T45 are two important chromatin modifications that accompany chromatin assembly. Here we have identified the protein kinase Pkc1 as a key regulator that coordinates the deposition of these modifications in *S. cerevisiae* under conditions of replicative stress. Pkc1 phosphorylates the histone acetyl transferase Rtt109 and promotes its ability to acetylate H3K56. Our data also reveal novel cross-talk between two different histone modifications as Pkc1 also enhances H3T45 phosphorylation and this modification is required for H3K56 acetylation. Our data therefore uncover an important role for Pkc1 in coordinating the deposition of two different histone modifications that are important for chromatin assembly.

**\*For correspondence:** andrew.d. sharrocks@manchester.ac.uk

**Competing interests:** The authors declare that no competing interests exist

## Introduction

During the cell cycle one of the key processes following DNA replication is the re-assembly of chromatin on the newly replicated DNA. This involves the re-deposition of both pre-existing and newly synthesised histones into intact nucleosomes (reviewed in *Groth et al., 2007*; *Ransom et al., 2010*; *Gurard-Levin et al., 2014*). This process must be tightly controlled and in *S. cerevisiae* one key event is the Rtt109-mediated modification of histone H3 by acetylation of lysine 56 (H3K56) (*Han et al., 2007*; *Driscoll et al., 2007*). Rtt109 cooperates with two different histone chaperones Asf1 and Vps75 (*Tsubota et al., 2007*; *Driscoll et al., 2007*) and although it is generally accepted that Asf1 is the major chaperone involved in promoting H3K56 acetylation (*Keck et al., 2011*; *Tang et al., 2011*), recent evidence suggests a role for Vps75 in this process (*Radovani et al., 2013*). However, the signalling pathways controlling Rtt109-mediated H3K56 acetylation are unknown. Another important modification that occurs coincidentally with DNA replication is H3 phosphorylation at threonine T45 (H3T45) but current evidence suggests that this modification has no relationship with the closely located H3K56 acetylation event (*Baker et al., 2010*). In the context of an unperturbed cell cycle, Cdc7-Dbf4 has been shown to be a major kinase involved in controlling H3T45 phosphorylation levels (*Baker et al., 2010*).

 *S. cerevisiae* possesses one protein kinase C isoform, Pkc1, and it is known to play a major role in cell wall integrity signalling (reviewed in *Levin, 2005*). Pkc1 has a growing number of nuclear functions and through genetic experiments this protein kinase has previously been implicated in cell cycle events (reviewed in *Levin, 2005*). Part of this role can be attributed to its connection to pathways sensing cell wall integrity and its effects on the activity of the downstream MAP kinase pathway. However MAP kinase-independent roles for Pkc1 have emerged and Pkc1 was recently shown to directly affect the cell cycle by inhibiting the activity of the transcriptional coactivator protein Ndd1 (*Darieva et al., 2012*). This finding suggested that Pkc1p might act as a checkpoint kinase under conditions of replicative stress where progression to G2 phase is inhibited. However, mutant strains containing *ndd1* mutant alleles lacking the Pkc1 phosphorylation sites did not fully re-capitulate the *pkc1* mutant phenotypes, indicating that additional defects are associated with loss of Pkc1

**eLife digest** Prior to cell division, DNA must be copied so that each new cell gets a complete copy of the cell's genetic instructions. But DNA is so long that it is stored in a heavily compacted form in the nucleus of the cell, with the strands of DNA coiled around several proteins called histones. Before the DNA is copied, it must be unfurled. Then each new copy of DNA must be repackaged to fit compactly inside the nucleus of each new cell.

If errors occur in the process of copying DNA, it can lead to genetic mutations that may cause diseases like cancer. To prevent this, cells have mechanisms to identify errors and correct them before the DNA is repackaged. This requires a pause to allow the repairs to occur before the DNA recoils. However, it is not completely clear how this process is controlled.

Now, Darieva et al. show that an enzyme called protein kinase C (or Pkc1 for short) is essential to repackaging DNA after the errors are corrected. Several experiments showed that Pkc1 plays an important role when cells were exposed to stressful conditions that potentially cause errors in DNA copying. Specifically, Pkc1 helps prepare the third histone protein (histone H3) so that DNA can recoil around it. Pkc1 waits until the stressful conditions have passed and the DNA has been repaired to make the necessary changes.

Once the stress has passed, Pkc1 adds a phosphate to another enzyme called Rtt109 that prepares the histone. The Pkc1 simultaneously contributes to another necessary change to histone H3. These new details about DNA repackaging may help researchers understand how cells protect against DNA copying errors, and how this process goes wrong in cancer.

activity. Indeed, *pkc1* mutants are sensitive to hydroxurea (HU) treatment (*Queralt and Igual, 2005*), suggesting a potential role for Pkc1 in controlling the response to replicative stress. It is unclear how Pkc1 might affect replicative stress but previous genetic studies have linked *PKC1* and components of the RSC chromatin remodelling complex and Pkc1 overexpression can suppress the defects associated with *RSC* mutants (*Chai et al., 2005*; *Hosotani et al., 2001*; reviewed in *Levin et al., 2005*). This is suggestive of a potential role for Pkc1 in regulating chromatin assembly during DNA replication.

Here, we have investigated the potential role of Pkc1 in controlling chromatin modifications during replicative stress and found that it coordinates the deposition of two histone marks on histone H3, T45 phosphorylation and K56 acetylation. Mechanistically, Pkc1 controls H3K56 acetylation through phosphorylation-dependent activation of Rtt109. Moreover, Pkc1 also promotes H3T45 phosphorylation and this modification is important for the coordinated deposition of H3K56 acetylation.

## Results

### Pkc1 is required for cell viability and chromatin integrity under replicative stress conditions

To examine whether Pkc1 has a role in protecting cells from replicative stress we first determined cell sensitivity to hydroxyurea (HU) in the presence and absence of active Pkc1. To do this we made use of strains containing temperature sensitive *pkc1* alleles which could be inactivated by incubating yeast cells at 34°C, thereby negating the confounding effects of heat shock stress response obtained using the traditionally used *pkc1^ts* allele (*Anastasia et al., 2012*). In comparison to cells containing wild-type *PKC1* (DK186), temperature sensitive *pkc1-14* (DK1690) cells, grown at 34°C to inactivate Pkc1 were inviable (*Figure 1A*, top panel) consistent with previous studies using a *pkc1Δ* deletion strain (*Queralt and Igual, 2005*). Loss of *PKC1* function results in cell lysis in the absence of osmotic support. The presence of sorbitol saved the *pkc1-14* mutant phenotype (*Figure 1A*, middle panel) but HU sensitivity was observed under these conditions (*Figure 1A*, bottom panel). This suggested a role for Pkc1 other than signalling through the well studied downstream MAP kinase cascade which includes the kinase Bck1 (*Lee et al., 1993*). Indeed *bck1Δ* (Y01328) cells were not sensitive to HU, further emphasising a new alternative mechanism of action for Pkc1 under replicative stress

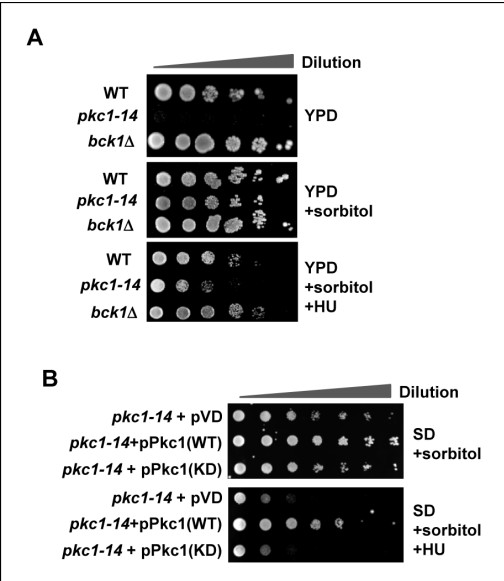

**Figure 1.** Pkc1p is required for cell viability and chromatin integrity in the presence of hydroxyurea. (**A** and **B**) Cell growth assays. (**A**) 10 fold serial dilutions of wild-type (WT) (DK186), *pkc1-14* (DK1690) or *bck1Δ* (Y01328) cells were plated onto YPD media in the presence or absence of 1M sorbitol or (**B**) *pkc1-14* (DK1690) cells containing empty vector or plasmids expressing WT or kinase dead (KD) Pkc1, were plated onto SD media in the presence 1M sorbitol. Cells were grown at 34°C and where indicated, 100 mM HU was added to the media.

conditions (*Figure 1A*). To determine whether the kinase activity of Pkc1 is needed to protect cells from replicative stress, we attempted to rescue *pkc1-14* cells by re-expressing a catalytically dead (KD) version of Pkc1 in the presence of HU. However, while wild-type (WT) Pkc1 was able to rescue the growth defect of *pkc1-14* cells, the catalytically dead version was unable to do so (*Figure 1B*).

Together, these experiments demonstrate that Pkc1 kinase activity is required for protecting cells from replicative stress.

## Pkc1 is required for H3K56 acetylation

During S phase, acetylation of histone H3 at lysine 56 (H3K56) is an important modification in ensuring the correct assembly of newly synthesised nucleosomes and their incorporation into chromatin (*Li et al., 2008*; *Masumoto et al., 2005*). We therefore examined whether Pkc1 is required for H3K56 acetylation under replicative stress conditions. Upon release from a HU block, wild-type cells showed accumulation of acetylated H3K56 (*Figure 2A*, lanes 1–6). In contrast, temperature sensitive *pkc1-21* cells (DK1697) showed much reduced levels of H3K56 acetylation when grown at 34°C (*Figure 2A*, lanes 7–12) even though they had progressed through S phase and arrested at G2 (*Figure 2—figure supplement 1A*). Importantly, this effect was not limited to strains harbouring the *pkc1-21* allele but was also observed in a different temperature sensitive strain, harbouring the *pkc1-14* allele either upon release from a HU block (*Figure 2B* and *Figure 2—figure supplement 1B*) or upon entry into S phase upon release from an α factor block (*Figure 2—figure supplement 2*). H3K56 acetylation is deposited by Rtt109, and this enzyme also acetylates histone H3 at lysine 9 (H3K9) (*Driscoll et al., 2007*; *Han et al., 2007*; *Tsubota et al., 2007*). Interestingly, in addition to H3K56 acetylation defects, *pkc1-14* cells showed much reduced levels of H3K9 acetylation (*Figure 2B*, middle panel) which suggests a defect in Rtt109 activity.

To further explore the role of Pkc1 in promoting H3K56 acetylation, we treated wild-type cells with the Pkc1 inhibitor cercosporamide (*Sussman et al., 2004*) and examined H3K56 acetylation following release from a HU block. Pharmacological inhibition of Pkc1 activity reduced H3K56 acetylation but importantly did not affect another histone H3 acetylation event at lysine 18 (H3K18) (*Figure 2C*), demonstrating the specificity of action of Pkc1. We also examined whether combinatorial genetic inactivation and pharmacological inhibition of Pkc1 has additive effects but none were

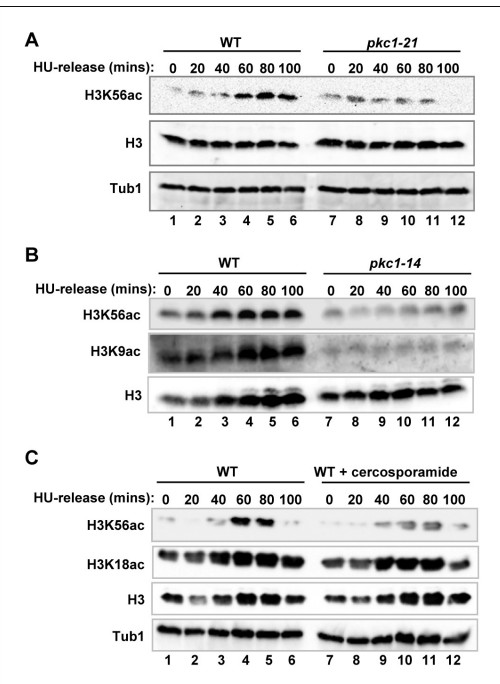

**Figure 2.** Pkc1p controls H3K56 acetylation. (**A-C**) Western blot analysis of the H3K56 acetylation levels in (**A**) WT (DK186) or *pkc1-21* (DK1697) or (**B**) WT (DK186) or *pkc1-14* (DK1690) strains grown in YPD in the presence 1M sorbitol at 34°C, treated with 200 mM HU for 2.5 hrs and released into YPD with sorbitol for the indicated times. (**C**) WT (DK186) cells grown in the presence or absence of cercosporamide at 30°C in YPD, exposed to 200 mM HU for 2.5 hrs and released from a HU block for the indicated times. H3K9 acetylation levels are also shown in (**B**) and Histone H3 and tubulin (Tub1) are shown as loading controls.

The following figure supplements are available for Figure 2:

**Figure supplement 1.** The *pkc1-21* and *pkc1-14* mutants exhibit a mitotic delay.

**Figure supplement 2.** The role of Pkc1 in controlling H3K56 acetylation.

**Figure supplement 3.** Comparison of genetic and pharmacological disruption of Pkc1 activity on H3K56 acetylation.

observed, indicating that the effect of cercosporamide was through Pkc1 (*Figure 2—figure supplement 3*). We note however that the loss of Pkc1 in the temperature sensitive strains resulted in stronger reductions in H3K56ac levels than following cercosporamide treatment, most likely due to incomplete Pkc1 inhibition in the latter case.

Together these results demonstrate an important role for Pkc1 in promoting H3K56 acetylation under replicative stress conditions.

## Pkc1 mediates Rtt109 phosphorylation

The reduced levels of H3K9 and H3K56 acetylation observed upon disruption of Pkc1 activity suggested that the acetyltransferase Rtt109 which deposits these modifications might be targeted by Pkc1. One mechanism might be through Pkc1-mediated control of Rtt109 phosphorylation. Inspection of Rtt109 sequence revealed a potential phosphorylation site at threonine 46 (T46) which partially matches the PKC consensus sequence (*Figure 3A*). To establish whether Rtt109 can be phosphorylated by Pkc1 at this site, we raised a phospho-specific antibody which specifically recognised a phosphorylated peptide surrounding T46 (*Figure 3—figure supplement 1A*). In vitro kinase assays revealed that Pkc1 was able to phosphorylate Rtt109 in the context of a purified recombinant Rtt109-Vps75 heterodimer (*Figure 3B*). This phosphorylation was reduced to background levels

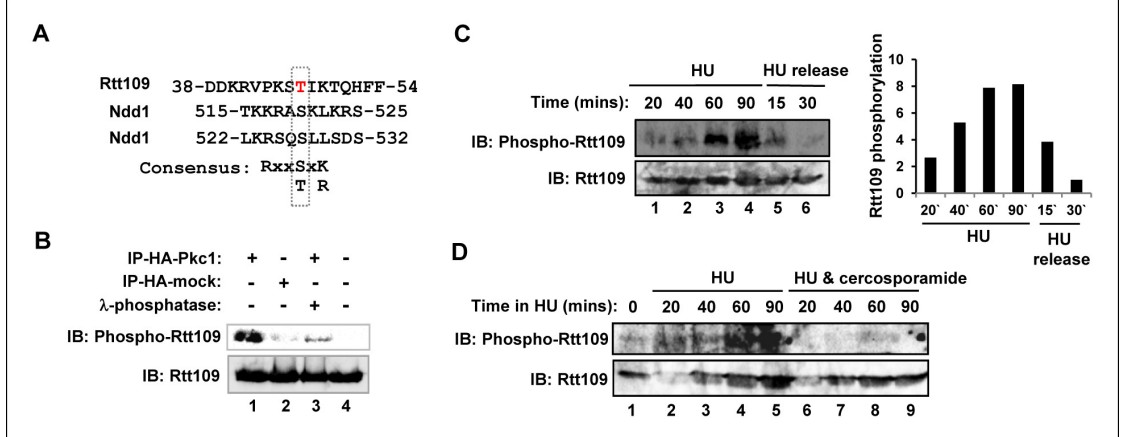

**Figure 3.** Pkc1-mediated Rtt109 phosphorylation. (**A**) Location and local sequence contexts of the potential Pkc1p target site in Rtt109 (numbers indicate amino acid positions in the protein). (**B**) In vitro kinase assay using HA epitope-tagged Pkc1p immunoprecipitated from DZ2 cells and purified recombinant Rtt109-Vps75 protein complex. A HA IP from wild-type yeast cells was used as a control. Where indicated, λ phosphatase was added. Rtt109, HA-Pkc1 and phosphorylated Rtt109 T46 (phospho-Rtt109) were detected by immunoblotting (IB). (**C** and **D**) Rtt109 T46 phosphorylation *in vivo*. DZ5 cells were grown in SD media at 30°C, treated with 200 mM HU for the indicated times and either (**C**) released from the HU block for the indicated times or (**D**) treated with or without cercosporamide. Rtt109 and phosphorylated Rtt109 T46 were detected by IB. Quantification of Rtt109 phosphorylation from (**C**) relative to total Rtt109 levels is shown on the right.

The following figure supplements are available for Figure 3:

**Figure supplement 1.** Pkc1 and Rtt109 phosphorylation.

following treatment with lambda phosphatase (*Figure 3B*, lane 3). Immunoprecipitation assays demonstrated that Pkc1 and Rtt109 could interact in vitro (*Figure 3—figure supplement 1B*).

Rtt109 phosphorylation at T46 could also be identified *in vivo* following HU treatment (*Figure 3C*, lanes 1–4). This phosphorylation was subsequently lost upon release from HU arrest (*Figure 3C*, lanes 5–6). Importantly, treatment of cells with cercosporamide prior to HU addition blocked the appearance of Rtt109 phosphorylation at T46 (*Figure 3D*), demonstrating a role for Pkc1 in promoting Rtt109 phosphorylation under replicative stress conditions.

These data therefore indicate that in cells undergoing replicative stress, Rtt109 is phosphorylated in a Pkc1-dependent manner, most likely through direct phosphorylation by Pkc1.

## Phosphorylation of Rtt109 is important for H3K56 acetylation

To establish whether phosphorylation of Rtt109 at T46 has any functional relevance, we attempted to rescue *rtt109Δ* cells lacking Rtt109 protein expression with plasmids encoding mutant proteins which had either lost the ability to be phosphorylated (T46A) or carried a phosphomimetic residue in place of T45 (T46D). In the presence of HU, *rtt109Δ* cells grew poorly (*Figure 4A*). This growth defect could be rescued by re-expressing wild-type Rtt109 but this rescue was lost with the non-phosphorylatable Rtt109(T46A) mutant. In contrast, Rtt109(T46D) was able to suppress the growth defect (*Figure 4A*) indicating that the insertion of a phosphomimetic residue at the Pkc1 phosphorylation site reinstated the activity of Rtt109. To understand the molecular basis to this rescue, we again examined H3K56 acetylation. As expected, this was severely depleted in *rtt109Δ* cells but restored upon re-expression of wild-type Rtt109 (*Figure 4B*, left panels). Rtt109(T46A) only partially re-constituted H3K56 acetylation levels and the kinetics were delayed whereas Rtt109(T46D) promoted rapid and high levels of H3K56 acetylation (*Figure 4B*, left panels). These results are fully consistent with a role for Pkc1-mediated phosphorylation of Rtt109 being required for its ability to promote H3K56 acetylation.

Interestingly, we found that while wild-type Rtt109 and Rtt109(T46D) were inducibly expressed to high levels upon release from HU arrest, Rtt109(T46A) expression was delayed and reached a lower level (*Figure 4B*, right hand panels). This suggested that Pkc1-mediated phosphorylation might be important for stabilising Rtt109. Indeed, we found that Rtt109 protein was expressed to lower levels

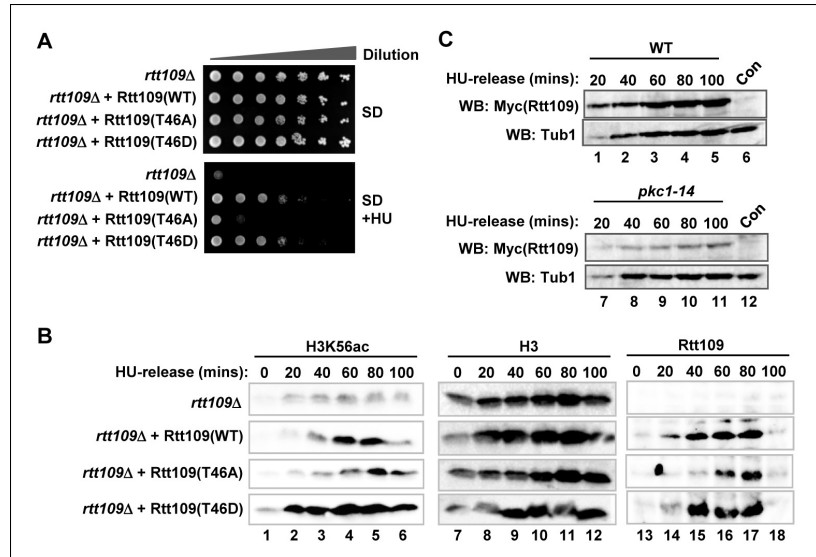

**Figure 4.** Rtt109 phosphorylation is needed for efficient H3K56 acetylation. (**A**) Cell growth assays. 10 fold serial dilutions of *rtt109Δ* (Y01490) cells containing plasmids expressing WT or the indicated Rtt109 mutants, were plated onto SD media in the presence or absence of 100 mM HU. (**B**) Western blot analysis of the H3K56 acetylation (ac) levels in *rtt109Δ* cells containing empty vector (DZ8) or plasmids expressing WT or the indicated Rtt109 mutants (DZ5-7) grown in SD media at 30°C and released from a 200 mM HU block for the indicated times. Total Rtt109 and Histone H3 levels are also shown. (**C**) Western blot analysis of Rtt109 levels in (WT) (DK186) or *pkc1-14* (DK1690) cells containing empty vector (con) or a plasmid containing Myc-tagged Rtt109, grown in SD media in the presence of 1 M sorbitol at 34°C and released from a 200 mM HU block for the indicated times.

The following figure supplements are available for Figure 4:

**Figure supplement 1.** *RTT109* expression is elevated in *pkc1-14* cells.

in *pkc1-14* cells (***Figure 4C***). This was not due to a transcriptional defect as *RTT109* mRNA levels were elevated in *pkc1-14* cells (***Figure 4—figure supplement 1***), suggesting that these cells are attempting to compensate for the loss in Rtt109 protein. This lower level of Rtt109 likely contributes to the decreased levels of H3K56 acetylation we observed. Together, these results indicated that Pkc1-mediated Rtt109 phosphorylation is required for maintaining Rtt109 protein levels and H3K56 acetylation levels as cells recover from HU-mediated replicative stress.

One prediction from these results is that any chromatin changes should manifest themselves in similar gene expression profiles when *pkc1-14* cells or cells expressing Rtt109(T46A) are compared. Indeed, in cells released from HU block, a highly significant overlap was observed for *pkc1-14* cells or cells expressing Rtt109(T46A) in genes either downregulated or upregulated compared to wild-type cells (***Figure 5A and B***). Importantly no such significant overlaps were observed when we considered genes that were regulated in opposite directions in the two mutant strains (***Figure 5B***). Furthermore, when we considered the biological processes that were affected, there was a strong overlap in gene ontology (GO) terms, with over-representation of GO terms associated with processes such as 'cell wall' and 'nucleolus' strongly associated with both mutant strains and reciprocal effects seen with 'telomere maintenance' and 'DNA helicase activity' (***Figure 5C and D***). These data therefore further support a strong association between Pkc1 activity, phosphorylation of Rtt109 at T46 and subsequent downstream effects on chromatin structure and gene expression. One prediction of these results is that the phosphomimetic version of Rtt109 might be able to bypass the defects seen in *pkc1-14* cells. However, no rescue of the growth defects of *pkc1-14* cells in the presence of HU was observed (***Figure 5—figure supplement 1***). Therefore a working model suggested a direct role for Pkc1 in controlling H3K56 acetylation through Rtt109 phosphorylation but also that additional activities were controlled by Pkc1 (***Figure 5E***).

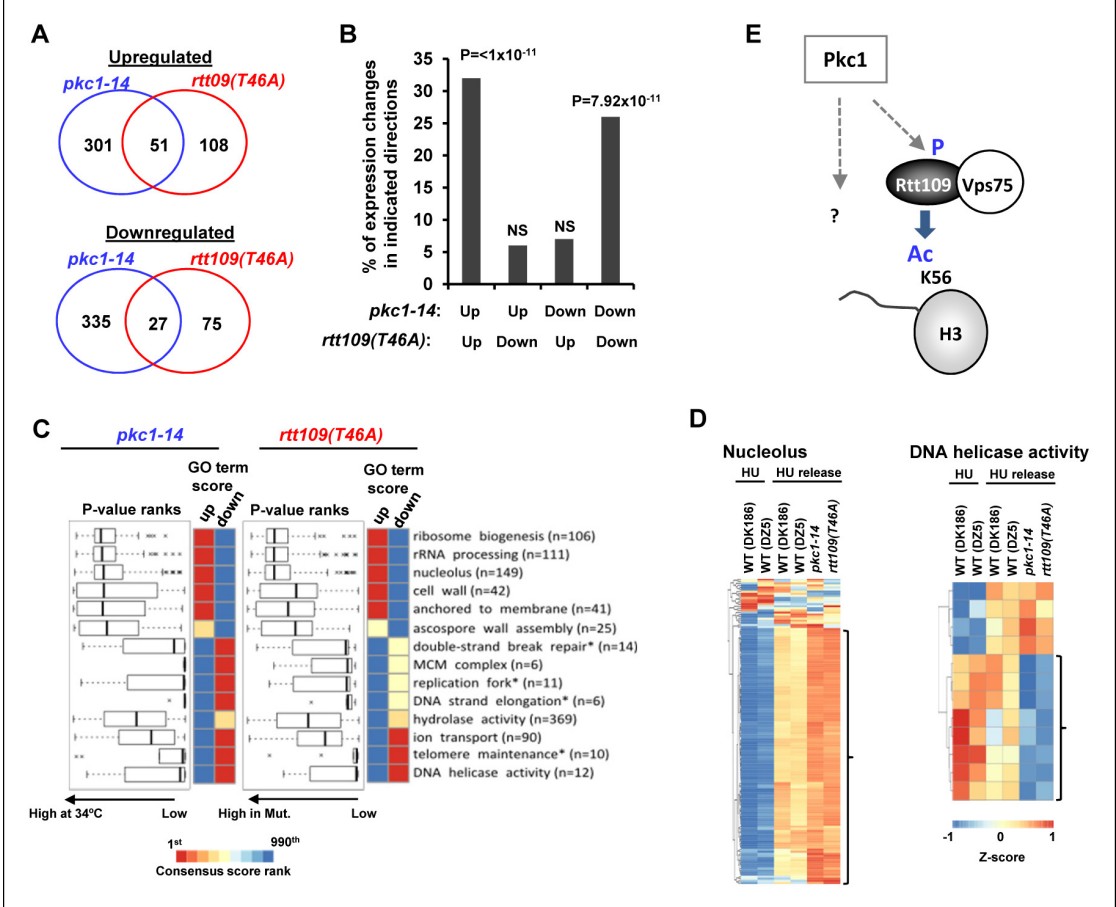

**Figure 5.** *Pkc1* and *RTT109* phosphorylation mutants generate overlapping gene expression defects. (**A**) Venn diagrams showing the overlap of significantly upregulated (top) or downregulated (bottom) genes in *pkc1-14* (DK1690) or *rtt109(T46A)* (DZ6) cells released from a 200 mM HU block. The experiment was performed by growing all strains in SD media supplemented with 1M sorbitol at 34°C. (**B**) Prevalence of gene expression changes showing the indicated directionality of change (shown as a percentage of genes changed in the *rtt109(T46A)* mutant strain). Hypergeometric P-values are shown (NS = non-significant). (**C**) Boxplots (left) and heatmaps (right) showing the rank orders of significantly changing GO terms associated with genes whose expression is consistently changed in the *pkc1-14* (DK1690) (left) or *rtt109(T46A)* (DZ6) (right) cells. The GO terms are ranked according to changes in the *rtt109(T46A)* (DZ6) mutant. In the heat maps, the columns indicate the P-value ranking of GO terms associated with genes that consistently increase (left) or decrease (right) their expression in the respective mutant strain. (**D**) Heat map showing the relative expression of the genes contained in the 'Nucleolus' and 'DNA helicase' categories across all conditions analysed. DK186 and DZ5 are the WT equivalent strains for the *pkc1-14* (DK1690) and *rtt109(T46A)* (DZ6) mutant strains. Brackets indicate genes that are consistently up- or down-regulated in both mutant strains upon release from a HU block. (**E**) Model for Pkc1 function through phosphorylating Rtt109 and promoting its ability to acetylate (Ac) histone H3 K56. Question mark indicates additional activities of Pkc1.

The following figure supplements are available for Figure 5:

**Figure supplement 1.** Rescue of the HU sensitivity of *pkc1-14* by a *rtt109* phosphomimetic allele.

## Pkc1 mediates histone H3 T45 phosphorylation

Previous results in mammalian cells had shown that PKCδ phosphorylates histone H3 at threonine 45 (H3T45) (*Hurd et al., 2009*). As this residue is conserved in *S. cerevisiae* (*Figure 6A*), and is located in the vicinity of K56, it is possible that Pkc1-mediated phosphorylation of T45 might influence acetylation events at K56. We therefore examined whether Pkc1 could phosphorylate H3T45 in vitro by using purified histone H3 and immunoprecipitated Pkc1p. Efficient H3 phosphorylation was observed (*Figure 6B*) and phosphorylation at T45 was confirmed as the only major phosphorylation site by mass spectrometry (*Figure 6C*). Next we examined H3T45 phosphorylation and its dependence on Pkc1 *in vivo*. As cells were released from a HU-mediated block, H3T45 became strongly

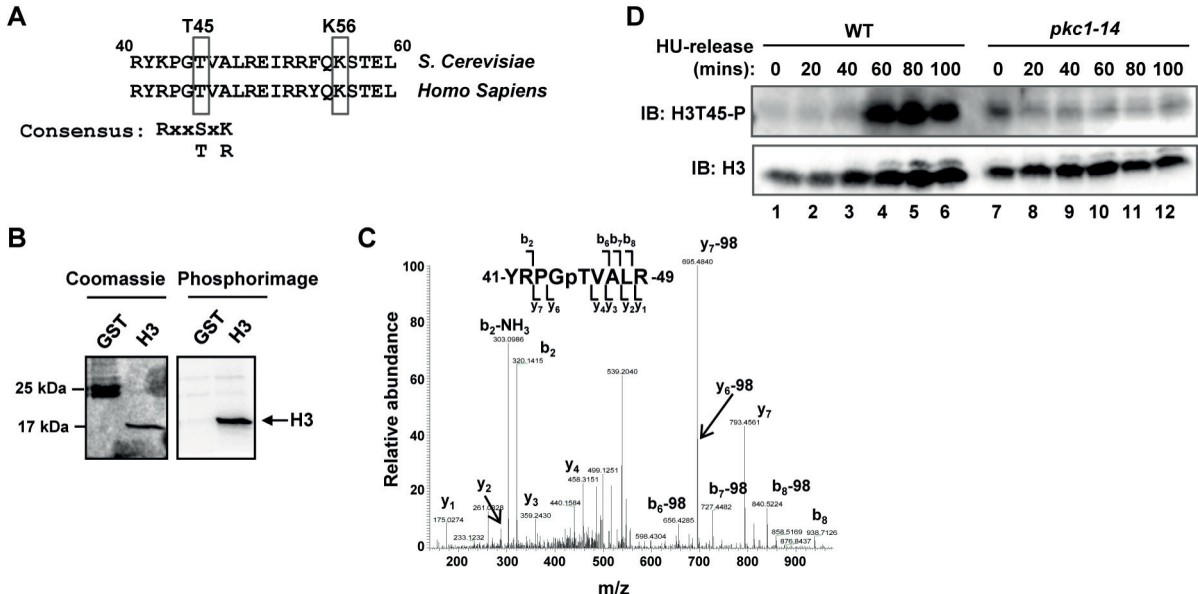

**Figure 6.** Pkc1 promotes histone H3 T45 phosphorylation. (**A**) Location and local sequence contexts of the Pkc1p target site in histone H3 (numbers indicate amino acid positions in the protein and human H3 is shown below). (**B**) In vitro kinase assay using HA epitope-tagged Pkc1p immunoprecipitated from DZ2 cells and recombinant histone H3 or GST as substrates. Protein phosphorylation was visualised by phosphorimaging (right panel) and total input proteins by Coomassie blue staining (left panel). (**C**) Product ion spectra for the doubly-charged precursor ion [556.78]²⁺. The spectra is fully annotated, and includes both b- and y-ions that support phosphorylation (**p**) at the residue indicated, Thr45. (**D**) Histone H3 T45 phosphorylation in vivo. WT (DK186) or pkc1-14 (DK1690) cells were grown in SD media in the presence of 1M sorbitol at 34°C and were treated with 200 mM HU and released from the HU block for the indicated times. Histone H3 and phosphorylated H3 T45 (H3T45-P) were detected by IB.

phosphorylated in wild-type cells (*Figure 6D*). However, no such rises in H3T45 phosphorylation were observed upon ablation of Pkc1 activity in *pkc1-14* cells (*Figure 6D*, lanes 7–12).

The results therefore indicate that Pkc1 mediates histone H3 phosphorylation under conditions of replicative stress.

## Pkc1-mediated histone H3 T45 phosphorylation promotes H3 K56 acetylation

Given the relatively close proximity of T45 to K56 in the histone H3 tail and the link of modifications of both of these residues to Pkc1 activity, we next asked whether there is a connection between H3T45 phosphorylation and H3K56 acetylation. Importantly, H3K56 acetylation was greatly reduced in cells containing the H3(T45A) allele upon release from a HU block (*Figure 7A*, lanes 3 and 4; *Figure 7B*). In contrast only a small change in H3K56 acetylation was observed upon release from an alpha factor block under normal growth conditions (*Figure 7A*, lanes 1 and 2). However, if cells were released from an alpha factor-mediated G1 arrest into HU, and the HU subsequently removed, H3K56 acetylation again accumulated following HU removal in wild-type cells. This accumulation of H3K56 acetylation did not occur in cells harbouring the *H3(T45A)* mutant allele (*Figure 7—figure supplement 1*). These results suggested functional coupling between H3T45 phosphorylation and H3K56 acetylation. To further probe this phenomenon, we asked whether the Rtt109(T46D) mutant could bypass the defects seen in cells containing the *H3(T45A)* mutant allele. However, only low levels of H3K56 acetylation were observed in the presence of Rtt109(T46D) (*Figure 7C*), demonstrating that this protein was unable to suppress the defects from H3(T45A) thereby indicating a crucial role for H3T45 phosphorylation in promoting H3K56 acetylation. Indeed, the Rtt109(T46D) mutant couldn't bypass the growth defects of *H3(T45A)* cells in HU (*Figure 7D*). We also tested whether Rtt109(D) could bypass the HU-mediated growth defects of *H3(K56R)* cells but again this mutant was unable to rescue the growth defect (*Figure 7—figure supplement 2*). This further emphasises

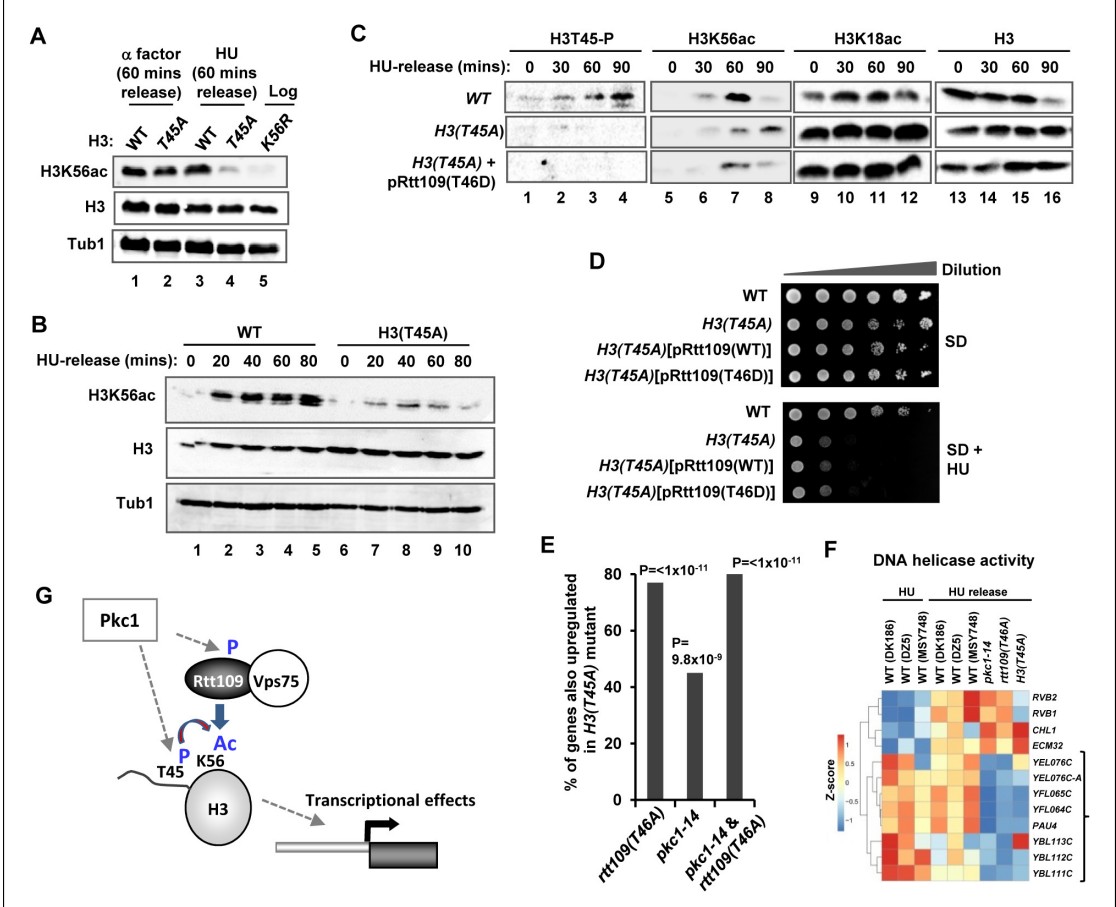

**Figure 7.** Phosphorylation of histone H3 T45 is required for efficient H3 K56 acetylation. (**A** and **B**) Western blot analysis of K56 acetylation (H3K56Ac) and total H3 levels in (MSY748) cells containing plasmids expressing H3(WT) or the H3(T45A) mutant following release from an alpha factor or HU block for the indicated times. Cells were grown in SD media at 30°C and treated with either alpha factor for 3 hrs or 200 mM HU for 2.5 hrs prior to release. Tubulin (Tub1) levels are shown as a loading control. Logarithmically growing (FXY19) cells containing a plasmid expressing the H3(K56R) mutant are shown as a control in (**A**). (**C**) Western blot analysis of T45 phosphorylation (H3T45-P), K56 acetylation (H3K56Ac), K18 (H3K18ac) and total H3 levels in cells containing plasmids expressing H3(WT) (MSY748) or the H3(T45A) (H3-T45A) mutant following release from a HU block for the indicated times. Where indicated cells also expressed Rtt109(T46D). (**D**) Cell growth assays. 10 fold serial dilutions of cells containing plasmids expressing H3(WT) (MSY748) or the H3(T45A) (H3-T45A) plus either Rtt109(WT) or Rtt109(T46D), were plated onto SD media in the presence (bottom panel) or absence (top panel) of 100 mM HU. (**E**) Frequency of gene expression changes upregulated in the *H3(T45A)* mutant strain that are also upregulated in either the *rtt109(T46A)* or the *pkc1-14* cells or in both of these mutant strains. Hypergeometric P-values are shown (NS = non-significant). (**F**) Heat map showing the relative expression of the genes contained in the 'DNA helicase' category across all conditions analysed. DK186, DZ5 and MSY748 are the WT equivalent strains for the *pkc1-14* (DK1690), *rtt109(T46A)* (DZ6) and *H3(T46A)* mutant strains respectively. The experiment was performed by growing all strains in SD media supplemented with 1M sorbitol at 34°C. Brackets indicate genes that are generally down-regulated in all of the mutant strains upon release from a HU block. (**G**) Model for Pkc1 function through phosphorylating Rtt109 and promoting its ability to acetylate (Ac) histone H3 K56 and acting through mediating H3 T45 phosphorylation which in turn enhances K56 acetylation.

The following figure supplements are available for Figure 7:

**Figure supplement 1.** H3K56 acetylation is lost in the *H3(T45A)* mutant strain upon release from an alpha factor block.

**Figure supplement 2.** Cell growth assays of strains containing histone H3 mutants in the presence of phospho-mimetic versions of Rtt109.

the importance of Rtt109-mediated H3K56 acetylation in protecting against the deleterious effects of HU treatment, even when H3T45 was available for phosphorylation. These results lead to a model whereby Pkc1 signals through promoting phosphorylation of both Rtt109 and histone H3 to promote H3K56 acetylation. One prediction from this model is that downstream gene expression defects should be at least partially shared by *pkc1-14*, *H3(T45A)* and *Rtt109(T46A)* mutant strains.

Indeed pairwise comparisons of the gene expression changes observed between these strains following release from a HU block, showed highly significant overlaps in genes being upregulated in all cases (*Figure 7E*; *Figure 5B*). Importantly though, of the 51 genes showing significant upregulation in the *pkc1* and *Rtt109(T46A)* mutant strains, 41 (80%) were also upregulated in the *H3(T45A)* mutant strain (*Figure 7E*). Moreover, consistent co-regulation of genes in the 'DNA helicase' GO term category could be observed across all mutant strains (*Figure 7F*). These mutants therefore exhibit common defects in gene expression. This finding provides further supporting evidence that adds to the mechanistic links we have uncovered between Pkc1, Rtt109, H3T45 phosphorylation and H3K56 acetylation, and strengthens our major discovery that Pkc1 coordinates histone H3 modifications during the replicative stress response.

## Discussion

Newly synthesised DNA must be efficiently packaged into chromatin to ensure that genome stability is maintained (reviewed in *Groth et al., 2007*; *Ransom et al., 2010*; *Gurard-Levin et al., 2014*). This must be achieved during both an unperturbed cell cycle and under conditions of replicative stress where the cell cycle must pause to ensure DNA repair occurs before recommencing the cycle. Histone H3 T45 phosphorylation and K56 acetylation are two important modifications that occur during S phase and are required for chromatin reassembly (*Masumoto et al., 2005*; *Hyland et al., 2005*; *Ozdemir et al., 2005*; *Li et al., 2008*; *Baker et al., 2010*). Here we show that in *S. cerevisiae* Pkc1 coordinates the deposition of these two marks under conditions of replicative stress. Previous results had implicated the Cdc7/Dbf4 kinase complex in directly controlling H3T45 phosphorylation during S phase of and unperturbed cell cycle but this kinase did not affect H3K56 acetylation (*Baker et al., 2010*). However, a role under replicative stress conditions was not investigated and residual levels of H3T45 phosphorylation were observed in the *cdc7Δ* cells, indicating the existence of another H3T45 kinase. Indeed, Cdc7/Dbf4 activity has been shown to be downregulated under conditions of replicative stress, suggesting the involvement of a different histone H3 kinase under these conditions (*Weinreich and Stillman, 1999*). Here we focussed on histone H3 modification control under replicative stress conditions and in combination with previous results (*Baker et al., 2010*) our data indicate that at least two different kinases are involved in controlling H3T45 phosphorylation, with Cdc7/Dbf4 operating during an unperturbed cell cycle and Pkc1 operating under replicative stress conditions where it also coordinates H3K56 acetylation.

In cells subjected to replicative stress it is important to delay chromatin re-assembly until the DNA damage has been resolved, and one way to do this, would be to delay H3 modifications. It is not clear how this delay is achieved but Pkc1 promotes the re-activation of H3 modifications upon removal of replicative stress. Mechanistically, Pkc1 controls H3K56 acetylation through phosphorylating the Rtt109 acetyltransferase. This modification stabilises Rtt109 and thus safeguards its availability for modifying histone H3 following removal of replicative stress and re-entry into the cell cycle. It is possible that Pkc1 might also directly target other elements of the chromatin modification machinery to coordinate efficient chromatin assembly such as occurs with Cdc7-Dbf4 which can also target CAF-1 (chromatin assembly factor 1) (*Gerard et al., 2006*) as well as H3T45.

Under replicative stress conditions, H3T45 phosphorylation appears to be critically important for coupled H3K56 acetylation, and this phosphorylation event is promoted by Pkc1-mediated H3T45 phosphorylation. This is likely direct as histone H3 is efficiently phosphorylated at this site in vitro. This leads to a model (*Figure 7G*) whereby Pkc1 promotes the coordinate modification of these two chromatin modifications and H3T45 phosphorylation is functionally coupled to H3K56 acetylation. It is not clear how this coupling is mediated but one possibility is that H3T45 phosphorylation blocks histone binding to DNA, thus allowing H3K56 to be exposed and hence accessible for acetylation by Rtt109. More complex scenarios are of course possible such as phosphorylated H3T45 acting as binding platform for recruiting a histone mark reader which then coordinates H3K56 acetylation. Indeed, further complexities in the regulatory links between histone H3 modifications and Pkc1-mediated signalling are suggested by the observation that Pkc1 levels appear to be reduced in *H3 (T45A)* mutant strains (data not shown). Future studies will be required to distinguish between and dissect out the underlying molecular mechanisms. One outcome of perturbed Pkc1 activity and its downstream molecular events following HU treatment is likely to be defective chromatin assembly and preliminary evidence suggest that this might be the case (data not shown). Our data show a

consistent sensitivity of *pkc1, rtt109(T46A)* and *H3(T45A)* mutants to HU treatment which complements previous data showing that *H3(K56A)* alleles are also sensitive to HU (*Matsubara et al., 2007*) and supports a role in replicative stress. However, the response to other stress inducing conditions such as MMS and 6-AU treatment is less clear. For example, the *H3(T45A)* strain is sensitive to both MMS and 6-AU whereas the *rtt109(T46A)* strain is only sensitive to MMS (data not shown). Thus different elements of the regulatory network we have uncovered may be used to respond to different stress inducers.

Another open question is how Pkc1 is specifically targeted/activated during replicative stress but one potential regulator would the cell cycle-dependent changes of the cell wall and signalling through the cell wall integrity pathway. Alternatively, other upstream pathways might be involved, and one potential link might be through TOR signalling as TORC2 has been shown to phosphorylate and activate Pkc1 (*Nomura and Inoue, 2015*) and TOR signalling has also been linked to H3K56 acetylation (*Chen et al., 2012*). Whether such a route operates during replicative stress or is employed under different conditions is not clear but re-emphasises the potential widespread importance of connections between Pkc1 and histone H3 modifications we have identified.

Our work may be directly relevant to mammalian systems where PKCδ has been shown to phosphorylate H3T45 under apoptotic conditions (*Hurd et al., 2009*), although others have identified AKT as an additional H3T45 kinase under DNA damage conditions (*Lee et al., 2015*). It is not clear whether PKC is linked to H3K56 acetylation in mammals but this modification is induced by replicative stress and its deposition is mediated by p300/CBP, the functional homologue of Rtt109 (*Das et al., 2009*; *Vempati et al., 2010*). Indeed, atypical protein kinase C zeta has been shown to phosphorylate and control the histone acetylation output of CBP in the context of neuronal differentiation (*Wang et al., 2010*). Given these links between PKC and CBP/p300 it appears likely that our findings will be directly relevant to human cancer where oncogene-driven replicative stress is an important contributory event (*Bartek et al., 2007*) and therefore warrants further investigation.

## Materials and methods

### Plasmid construction and mutagenesis

For bacterial expression: pET28a+His-Rtt109 and pET3a-Tr-Vps75-Flag was kindly provided by Paul Kaufman (*Tsubota et al., 2007*).

For yeast expression; pVD67 (pAS1995; encoding wild-type GFP-Pkc1), pVD61(empty vector) and pVD124 (kinase-dead GFP-Pkc1) were kindly provided by Martha Cyert (*Denis and Cyert, 2005*). pAS4104 encoding Myc-Rtt109, pAS4105 (myc-Rtt109T46A) and pAS4106 (myc-Rtt109T46D) were made by inserting BamHI-cleaved PCR product generated using the templates pAS4101-4103 and primer pair ADS4723/4724 into pCM188. pAS4101 was generated by amplifying the cDNA sequence of the *RTT109* gene encoding full-length protein from yeast genomic DNA and cloning into pGBKT7 using EcoRI/SalI restriction sites. pAS4102 (encoding Rtt109[T45A]) and pAS4103 (encoding Rtt109[T45D]) were created by QuikChange mutagenesis (Stratagene) using the primer pairs ADS4727/ADS4728, ADS4729/4730 and pAS4101 as a template. pGal1Vps75-flag was kindly provided by Tom Owen-Hughes.

### Protein production, pulldown assay and western blotting

His-tagged fusion proteins were prepared using Ni-NTA agarose beads (Qiagen, UK) according to the manufacturer's protocol and HA epitope-tagged Pkc1p were prepared essentially as described previously (*Darieva et al., 2012*). Recombinant histone H3 was made commercially (Active Motif).

Pulldown assays with immunoprecipitated HA-Pkc1 from DZ2 cells and in vitro translated Rtt109 were carried out as described previously (*Pic-Taylor et al., 2004*).

To detect epitope-tagged derivatives by western analysis, anti-HA (Roche) and anti-tubulin TAT-1 (CRUK), H3K56ac (Active motif), H3K9ac (Abcam), H3K18ac (Abcam), H3 (Active motif), myc epitope (Santa Cruz), H3T45-P (Active motif) antibodies and Supersignal west dura substrate (Pierce) were used.

Rabbit polyclonal monospecific antibody against the phosphothreonine at amino acid 46 on yeast Rtt109 (Rtt109 T46-P) was generated from immunizing rabbits with a KLH-conjugated peptide (H-D-DKRVPKST(PO$_3$H$_2$)IKTC-NH2), and then cross-affinity purification of polyclonal antibodies specific to

modified Rtt109 T46-P or non-modified pRtt109 was performed (Eurogentec). For immunoblotting analysis we used 1:1000 dilution for both antibodies.

## Protein kinase assays

Protein kinase assays were performed as described previously (*Darieva et al., 2012*) using HA epitope-tagged Pkc1p immunoprecipitated form DZ2 cells and recombinant H3 or His-Rtt109/Vps75 protein complex as a substrate.

## Mass spectrometry

Proteins were isolated from denaturing polyacrylamide gels and proteolytically digested as described previously (*Darieva et al., 2012*). Digested samples were analysed by LC-MS/MS using an UltiMate® 3000 Rapid Separation LC (RSLC, Dionex Corporation, Sunnyvale, CA) coupled to an Orbitrap Elite (Thermo Fisher Scientific, Waltham, MA) mass spectrometer. Peptide mixtures were separated using a gradient from 92% A (0.1% FA in water) and 8% B (0.1% FA in acetonitrile) to 33% B, in 44 min at 300 nl min$^{-1}$, using a 75 mm x 250 µm i.d. 1.7 µM BEH C18, analytical column (Waters). Peptides were selected for fragmentation automatically by data dependant analysis. Data were analysed as described previously (*Darieva et al., 2012*) but phosphorylated peptide product ion spectra were also manually validated.

## Yeast growth and RNA analysis

The yeast strains used are listed in *Supplementary file 1*. Yeast cells were grown, *GAL1*-promoter-driven constructs induced, and transformations performed, as described previously (*Darieva et al., 2003*). Yeast cultures were treated with 40 nM cercosporamide (Santa Cruz) to inhibit PKC activity, synchronised in G1 phase by treatment with α-factor and in early S phase by hydroxyurea (HU) treatment as described previously (*Darieva et al., 2012*). DNA content analyses were performed using propidium iodide-stained cells as described previously (*Darieva et al., 2006*).

Growth sensitivity tests were performed as described previously (*Darieva et al., 2012*) except that temperature sensitive *pkc* mutant strains were grown at 30°C or 34°C. The experiments were performed in duplicate and were repeated several times.

RNA extraction and real-time reverse transcription-PCR (RT-PCR) analysis were carried out as described previously (*Darieva et al., 2006*) using the following primer pairs: ADS1587/1588 (*HTT2*), ADS4732/4733 (*RTT109*) and ADS3978-ADS3979 (*18S rRNA*). All data were normalized to *18S rRNA* levels in the same cells.

## Microarray analysis

Duplicate RNA samples were labelled and hybridised after a single round of amplification to Affymetrix Yeast Genome 2.0 array chips. All the cell strains used in this experiment were grown in the presence of 1M sorbitol at 30°C, treated with 200 mM HU for 2.5 hr and released from HU block at 34°C. Samples were obtained from: (1) Wild-type (DK186) and *pkc1-14* (DK1690) mutant strains. The mRNA expression data were collected at three timepoints (wild-type blocked in HU, wild-type and *pkc1-14* released from HU block for 75 min. (2) *rtt109Δ* mutant cells (Y01490) containing vectors expressing wild-type (WT) Rtt109 (DZ5) or Rtt109(T46A) (DZ6). The mRNA expression data were collected at three timepoints (WT blocked in HU, and Rtt109(WT) and Rtt109(T46A) released from HU block for 80 min. (3) Wild-type cells (MSY748) expressing wild-type histone H3 (MSY748) or H3 (T45A) (H3-T45A). The mRNA expression data were collected at three timepoints (WT blocked in HU, and both WT and H3(T45A) released from HU block for 80 min). The measurements were then normalised by using robust multi-array average (RMA) approach (*Irizarry et al., 2003*). The primary data are deposited in ArrayExpress; accession number E-MTAB-3359.

Limma (*Smyth, 2005*) was used to provide log(foldChange), T-statistic and P-values. In all cases, fold changes were calculated relative to the wild-type strain released from HU block. Individual gene expression changes were considered significant if they satisfied the criteria of fold change >1.2 and P-value <0.05. The T-statistics, P-values and Fold Changes were used by Piano (*Väremo et al., 2013*) to build aggregate P-values for all of the genes in each GO term, based upon the values of these statistics for each of the probe sets matching to the term. GO terms were then ranked

separately by aggregate P-values, within each of the five P-value directional classes (upregulated or downregulated in both conditions), and this ranking provided a consensus score.

To create heatmaps of expression changes of genes within GO terms z-scores were calculated for the set of arrays testing a pair of strains across different conditions and an average z-score for each condition determined. Each paired experiment was treated independently and the z-scores were clustered and depicted in a heatmap using pheatmap (http://cran.r-project.org/web/packages/pheatmap/index.html).

## Acknowledgements

We are grateful to Karren Palmer for technical support, and to staff in the core genomics technologies, mass spectrometry, bioinformatics, and flow cytometry facilities. We also thank Catherine Millar, Alan Whitmarsh, Iain Hagan and members of our laboratories for comments on the manuscript and helpful discussions, and Catherine Millar, Martha Cyert, Paul Kaufman, Tom Owen-Hughes, and Doug Kellog for reagents. This work was supported by the BBSRC, the Wellcome Trust and a Royal Society-Wolfson award to ADS.

## Additional information

### Funding

| Funder | Grant reference number | Author |
|---|---|---|
| Biotechnology and Biological Sciences Research Council | BB/H1010858/1 | Andrew D Sharrocks |
| Wellcome Trust | 103857/Z/14/Z | Andrew D Sharrocks |
| Royal Society Wolfson | | Andrew D Sharrocks |

The funders had no role in study design, data collection and interpretation, or the decision to submit the work for publication.

### Author contributions

ZD, ADS, Conception and design, Acquisition of data, Analysis and interpretation of data, Drafting or revising the article; AW, Conception and design, Analysis and interpretation of data, Drafting or revising the article; SW, Acquisition of data, Analysis and interpretation of data, Drafting or revising the article

## Additional files

### Supplementary files

• Supplementary fle 1. Yeast strains used in this study.

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
