## [Decision Letter]

Thank you for submitting your work entitled "Protein kinase C coordinates histone H3 phosphorylation and acetylation" for peer review at *eLife*. Your submission has been favorably evaluated by Jim Kadonaga (Senior editor), and two reviewers, one of whom is a member of our Board of Reviewing Editors.

The reviewers have discussed the reviews with one another and the Reviewing editor has drafted this decision to help you prepare a revised submission. The reviewers felt that your work was interesting, important and timely.

Essential revisions:

1) All MNase analyses have to be removed from the paper.

2) Show the effect on H3 K56Ac of Pkc inhibition in the *pkc1-14* and *pkc1-21* mutants, to establish whether they are epistatic, given their different effect on the kinetics of H3 K56Ac.

3) Indicate concentrations of HU and temperatures used for all experiments.

4) Include H3K56A or R mutants in the epistasis analysis in Figure 7 to enable a better understanding of the codependence of H3 K56Ac, H3 T64p and Rtt109p.

5) Make the gene expression data publicly available.

6) Please show or report result of sensitivities of the *pkc1*, H3T45 and Rtt109T46 alleles to MMS and 6AU.

We have included the full reviews for your benefit below.

*Reviewer #1:* This is an interesting paper that shows that Pkc1 phosphorylates the HAT Rtt109 to stabilize Rtt109 after release from HU treatment, enabling it to acetylate H3 K56 during S phase. They also show that Pkc1 promotes phosphorylation of H3 T45, which also promotes H3 K56Ac. The findings are novel and important. Some of the westerns are rather ugly but are still convincing. The micrococcal nuclease analyses however are not technically acceptable nor credible, nor consistent with data published for the same mutants in the literature previously. Additionally, they do not add anything to the story and need to be removed, given the effect of lack of H3 K56Ac on chromatin and MNase ladders is well established already.

For future reference this is what is wrong with the MNase ladders (probably irrelevant here, because these all have to be removed from the paper), MNase accessibility analysis requires multiple time points or different MNase concentrations to be shown for each chromatin sample, not just one. The gel percentages and ladders need to be used that will enable 150bp and 300bp fragments to be seen. Also, the same amount of sample has to be digested for different samples. This is clearly not the case in these MNase analyses, especially Figure 2 and Figure 5. For the MNase digestion in Figure 1, the wild type sample is totally unconvincing. It is not a MNase ladder – there are intermediate bands between the nucleosomes and the tri-nucleosome is far brighter than the mono and di bands, but there is no tetra. This does not happen; they never look like this; and the state of overdigestion of the Pkc1 mutant would not be compatible with life! The results of the MNase digestion analysis of H3 K56R is far more extreme than what Steve Jackson published in Nature nearly ten years ago, and this degree of relative lack of chromatin would not support life. Figure 2 MNase also lacks a ladder. The labeling of the size markers on Figure 5 does not correspond to the size markers and appears to be completely arbitrary. Where is the mononucleosome on this analysis? A ladder cannot have di and tri and no mononucleosome.

*Reviewer #2:* Sharrocks and colleagues present a number of interesting experiments that implicate the yeast PKC kinase in chromatin assembly regulation. Yeast growth phenotypes are applied to implicate pkc1 alleles in HU resistance and H3K56 acetylation as well as to study H3 and Rtt109 PKC phosphorylation site mutant alleles. Co-IP and in vitro and in vivo phosphorylation assays suggest a network of phosphorylation whereby PKC may directly phosphorylate H3 on T45 and Rtt109 on T46. The emergent model is attractive because there appears to be a two-pronged PKC-mediated regulatory pathway, whereby direct PKC phosphorylation of H3T45 promotes acetylation of H3K56 whilst concomitant phosphorylation of the H3K56 acetylase Rtt109 on T46 also promotes H3K56 acetylation. The model appears to have been tested adequately to sustain the claims, including epistasis experiments using colony formation and gene expression as a read-outs.

The MNase experiments are scattered over three figures (1C, 2C, 6E), assessing MNase patterns in wild type, *pkc1-14* (1C) H3K56R (2C) and a H3T45A mutant +/- PKC1 expression (6E). It is not clear whether as such these add substantially to the manuscript. Technically a full titration/time course of MNase for each strain/condition would be more informative. Furthermore, as different strain backgrounds are compared (1C = W303, 2C = Rothstein 1983 = W303, 6E = FY406), at different temperatures (30ºC? and 34ºC), the only result is that the chromatin structure is MNase hypersensitive and therefore 'messed up'. The assays displayed do not have the resolution to indicate in what way or to what extent the chromatin is messed up. Darieva et al. again do not systematically indicate at what temperature the control strain was grown. I feel that one dedicated accessibility figure describing a logical set, rather than the present chronological-discovery-set across three figures would render the manuscript stronger on this front. For instance, comparing WT, H3T45A and H3T45D combined with *pkc* mutants at 34ºC. A further set of possibilities would include the H3K56A, and R mutations. Yet another, would employ the rtt109 null and Rtt109A and Rtt109D alleles. Leaving MNase experiments out altogether is also an option, however.

---

## [Author Response]

*1) All MNase analyses have to be removed from the paper.*

We have now removed all of this data and amended the text appropriately. We would like to emphasise however that the MNase digests were carried out on chromatin isolated from cells released from HU treatment (perhaps not made clear in the paper). Under these conditions, it might be expected that chromatin would not be fully assembled correctly as many cells are still in S phase. In the case of defective Pkc signalling (and its downstream molecular events) we consistently saw changes relative to WT cells, and the increased MNase susceptibility would likely be resolved after longer release times from HU block (and hence compatible with cell survival). We do however appreciate that there are technical shortcomings with the current data (although we did use equal amounts of chromatin for each digestion) and agree that the data are not central to the paper and merit further investigation to pin down the precise defects involved. We have now mentioned this possibility in the Discussion as an area to investigate more fully in the future.

*2) Show the effect on H3 K56Ac of Pkc inhibition in the* pkc1-14 *and* pkc1-21 *mutants, to establish whether they are epistatic, given their different effect on the kinetics of H3 K56Ac.*

We have done this experiment and included it in the revised version (see Figure 2—figure supplement 3). This clearly demonstrates that the Pkc1 inhibitor (cercosporamide) inhibits H3K56Ac in wild-type cells but does not further affect the residual levels of H3K56Ac in cells depleted of Pkc1 activity in the *pkc1-14* temperature sensitive mutant strain. The difference in effects is likely caused by incomplete inhibition of Pkc1 activity by the inhibitor, leading to less severe effects on H3K56Ac levels (now commented on in the text).

*3) Indicate concentrations of HU and temperatures used for all experiments.*

This information has now been added to the figure legends where currently missing. We have also clarified the conditions of the microarray experiments in the Materials and methods and figures (which were not labelled properly in the previous version of the paper) and legend.

4) Include H3K56A or R mutants in the epistasis analysis in Figure 7 to enable a better understanding of the codependence of H3 K56Ac, H3 T64p and Rtt109p.

These additional experiments has now been done (new Figure 7—figure supplement 2) and demonstrate that Rtt109(T46D) is unable to rescue the defects in the H3K56R mutant strain. This lends further support for the model that Rtt109 functions primarily through H3K56Ac in this context.

*5) Make the gene expression data publicly available.*

This data was already deposited in the ArrayExpress database (mentioned in first version of the manuscript) and will be released to the public upon publication.

*6) Please show or report result of sensitivities of the* pkc1*, H3T45 and Rtt109T46 alleles to MMS and 6AU.*

We have now performed these experiments and there is a complex behaviour which is beyond the scope of this paper to investigate in detail. We have therefore not shown this data in the current paper. However, the data do show that although the phenotypic defects we see for all the mutant alleles investigated in this study under HU-mediated stress conditions, show consistent sensitivity responses, this is not the case in the context of MMS or 6-AU treatment which are known to elicit different molecular defects in yeast cells. We have however included an additional paragraph in the Discussion section to highlight this point:

**“**Our data show a consistent sensitivity of *pkc1, rtt109(T46A)* and *H3(T45A)* mutants to HU treatment which complements previous data showing that *H3(K56A)* alleles are also sensitive to HU (Matsubara et al., 2007) and supports a role in replicative stress. However, the response to other stress inducing conditions such as MMS and 6-AU treatment is less clear. For example, the *H3(T45A)* strain is sensitive to both MMS and 6-AU whereas the *rtt109(T46A)* strain is only sensitive to MMS (data not shown). Thus different elements of the regulatory network we have uncovered may be used to respond to different stress inducers.”